# Development of Geopolymer-Based Materials with Ceramic Waste for Artistic and Restoration Applications

**DOI:** 10.3390/ma15238600

**Published:** 2022-12-02

**Authors:** Laura Ricciotti, Alessio Occhicone, Claudio Ferone, Raffaele Cioffi, Oreste Tarallo, Giuseppina Roviello

**Affiliations:** 1Department of Architecture and Industrial Design, University of Campania, Luigi Vanvitelli, 81031 Aversa, Italy; 2Department of Engineering, University of Naples ‘Parthenope’, Centro Direzionale, Isola C4, 80143 Napoli, Italy; 3INSTM Research Group Napoli Parthenope, National Consortium for Science and Technology of Materials, Via G. Giusti, 9, 50121 Firenze, Italy; 4Department of Chemical Sciences, University of Naples Federico II, 80126 Naples, Italy

**Keywords:** geopolymer, ceramic wastes, mortars, art and design, recycling

## Abstract

This contribution presents the preparation and characterization of new geopolymer-based mortars obtained from recycling waste deriving from the production process and the “end-of-life” of porcelain stoneware products. Structural, morphological, and mechanical studies carried out on different kinds of mortars prepared by using several types of by-products (i.e., pressed burnt and extruded ceramic waste, raw pressed and gypsum resulting from exhausted moulds) point out that these systems can be easily cast, also in complex shapes, and show a more consistent microstructure with respect to the geopolymer paste, with a reduced amount of microcracks. Moreover, the excellent adhesion of these materials to common substrates such as pottery and earthenware, even for an elevated concentration of filler, suggests their use in the field of technical-artistic value-added applications, such as restoration, conservation, and/or rehabilitation of historic monuments, or simply as materials for building revetments. For all these reasons, the proposed materials could represent valuable candidates to try to overcome some problems experienced in the cultural heritage sector concerning the selection of environmentally friendly materials that simultaneously meet art and design technical requirements.

## 1. Introduction

Ceramic products embody a fundamental segment of the Italian manufacturing industry: currently, this business, which includes the production of sanitary ceramics, ceramic tiles and slabs, porcelain and tableware, technical ceramics and bricks, and refractory materials, includes 279 companies and over 27,500 employees with a turnover of 6.5 billion euros [1]. In particular, porcelain stoneware represents the main type of production in the Italian ceramic industry: 400 million square metres were manufactured in 2019, accounting for about 80% of the total production [1].

Porcelain stoneware is composed of illitic kaolinitic clays, sodium-potassium feldspars, and feldspar sands as the predominant parts and chromophoric oxides (usually iron and titanium oxides) as the minor part [2]. It is characterized by very low porosity and presents remarkable technological properties such as high mechanical, chemical, abrasion, and stain resistance [3]. These features make it an ideal material for applications in several areas, from building components such as floors to producing artistic objects.

In terms of environmental impacts, the main factors affecting the production of porcelain stoneware are polluting emissions, the extraction of raw materials and their transport (lead, fluorine, boron, powders, CO_2_) [4,5], the consistent consumption of energy (mainly consumption of methane gas), water, and the production of solid wastes that usually are destined for landfills [4,5]. Indeed, it has been estimated that about 15 to 30% of this production is considered a by-product and/or waste generally intended for landfills.

Recently, ceramic products and their by-products have been used as fillers in cementing materials for improving their durability and mechanical performances, thanks to the presence of silicoaluminate crystalline components [6]. However, the reuse of ceramic scraps in the building field is rather negligible and in the early stages of its diffusion [7].

The research for alternative sustainable materials has gained great attention due to the green policy introduced by the European Union through the introduction of the Next Generation EU package [8], based on the circular economy and the Italian Ecological Transition Plan, and has created effective routes for the reuse and valorization of a massive amount of industrial wastes.

In this context, while the design of “green” products with a low carbon footprint and a reduced environmental impact according to the principles of the “eco-design” is well structured in sectors such as building and construction [9,10,11,12,13,14], to date, several issues have been pointed out in the case of the creative and restoring industry, particularly in terms of using of sustainable materials to meet eco-design features [15]. Traditionally, in fact, this sector has only been concerned with providing aesthetic enhancements by using an approach that ignores energy savings and emissions reduction [15]. However, in more recent years the approach has completely shifted toward the so-called eco-design, a new branch of design that aims to integrate environmental aspects during the process of designing products as any other criterion (economic, technological, and so on), to reduce their life cycle impacts. In order to reduce environmental impacts, its directives should be more enforced and routinely integrated into the product development process.

In this regard, geopolymers and alkali-activated materials, which can be made from widely available minerals, are perfect candidates to produce materials with low environmental impacts. In fact, aluminosilicate materials, which are major components of the Earth’s crust (65%) [16] as virgin materials or by-products of other industrial processes, are used as raw materials for the synthesis of geopolymer materials [17].

These raw materials can react in alkaline conditions and form amorphous species, characterized by cross-linked networks consisting of Si–O–Al–O bonds [18], that are considered a valid alternative to cement-based materials due to their characteristics such as thermal stability, low shrinkage, freeze-thaw, chemical and fire resistance, recyclability, and long-term durability [19].

One of the most interesting aspects is the possibility of preparing geopolymers by valorizing waste materials from different types of processing. Moreover, it is well known that aluminosilicate materials from bottom ash deriving from blast furnace slag, fly ash from thermoelectric industry residues, red mud, lake residues, and up to rice husk waste can be profitably exploited, giving life to new materials that find applications in various fields, from construction, to the manufacturing of objects, to the development of innovative materials for technological applications, when they are functionalized or produced as hybrids or composites [20,21,22].

In this regard, the authors of this paper have contributed over the years to this research by developing composites and hybrid materials based on geopolymers able to overcome some limitations of geopolymers that perimeter their extensive application in the construction sector, thus reaching the development of new materials with excellent mechanical performance and good thermal-acoustic insulation properties [23,24,25,26,27,28,29,30,31,32,33,34,35,36,37,38].

To the best of our knowledge, the excellent mechanical properties, fast setting time, easy workability, and ability to adhere to different types of substrates of geopolymer-based materials to date were exploited mainly in the construction sector, while no attention was devoted to the application of these materials in the artistic and/or restoration sectors. As a matter of fact, the previously listed properties, combined with the high chemical resistance and good mechanical performances of geopolymer materials, make them excellent candidates for the restoration field and artistic industry.

In this work, geopolymer-based mortars obtained through the valorization of wastes deriving from the production and “end-of-life” of porcelain stoneware products are proposed as eco-friendly materials to be used in the art and design sector. The objectives of this study have been both investigating the potential use of large amounts of ceramic waste in geopolymer-based mortars, since it is an interesting option for ceramic industries and today more and more are interested in recycling and producing sustainable materials, and, at the same time, suggesting the exploitation of these materials in the artistic and/or restoration sectors.

Different kinds of geopolymer mortars were prepared by using all kinds of by-products of the production process of the porcelain stoneware (pressed burnt and extruded ceramic wastes, raw pressed and gypsum resulting from exhausted moulds). These wastes were physically and chemically characterized, and then used as a recycled aggregate.

It is suggested that such materials can be used in the field of the art and design industry since they have excellent rheological properties, are suitable for casting in moulds with complex shapes, and, once consolidated, show good mechanical properties. Moreover, they do not show appreciable shrinkage, have a water absorption capacity similar to unmodified geopolymers, and show an excellent adhesive capacity to various types of substrates such as ceramic, earthenware, tuff, concrete, and marble, even for an elevated concentration of filler, thus suggesting that they could be used also for restoration and consolidation of artistic or archaeological artefacts.

The materials developed in this work potentially offer the possibility of recovering and recycling up to 100% (almost endlessly) of not only waste materials and by-products of the ceramic industries (such as gypsum, raw pressed, and porcelain stoneware waste), but also the geopolymers themselves at the end of their use. In this way, a sensible reduction of the environmental impacts of these materials could be achieved and, in line with the Circular Economy approach, green and economically competitive products could be obtained.

The possibility of extending this approach to valorize and recycle ceramic wastes from different kinds of industries, together with in-depth structural and durability tests and studies on advanced modelling of the architectural composition, could open up new scenarios for making structural and functional elements for sustainable and advanced buildings.

## 2. Experimental Section

### 2.1. Materials

Metakaolin MetaMax^®^ (by BASF) was kindly provided by Neuvendis s.p.a. (Milan, Italy) and its composition is reported in Table 1. BASF MetaMax^®^ is a high-purity white mineral admixture that meets or exceeds all the specifications of ASTM C-618 Class N pozzolans. The sodium silicate solution (Table 1) was supplied by Prochin Italia S.r.l (Caserta, Italy), while sodium hydroxide with reagent grade was supplied by Sigma-Aldrich. Ceramic waste and scraps, selected from those produced in greater quantities in the various stages of the porcelain manufacturing process (pressed burnt and extruded ceramic waste, raw pressed and gypsum resulting from exhausted moulds), were kindly supplied by a company producing stoneware and ceramic tiles in the province of Vicenza, Northern Italy.

### 2.2. Sample Preparation

#### 2.2.1. Geopolymer (MK)

It is well known that the geopolymerization reaction is based on the alkaline activation of an aluminosilicate raw material using a strongly alkaline solution. In this work, NaOH in pellets was dissolved in the sodium silicate solution to prepare the alkaline activating solution.

The solution so prepared was cooled and allowed to equilibrate for 24 h, as reported in refs. [30,31,32,33,34,35,36,37,38]. The chemical composition of the alkaline activating solution is Na_2_O 1.55 SiO_2_ 12.14 H_2_O. BASF Matamax^®^ metakaolin was then incorporated into the activating solution with a liquid-to-solid ratio of 1.4:1 by weight and mixed by a mechanical mixer for 10 min at 800 rpm [25,26,27,28,29,30]. The composition of the whole geopolymer system can be expressed as Al_2_O_3_ 3.48 SiO_2_ 1.0 Na_2_O 12.14 H_2_O, as revealed by EDS analysis carried out on the cured samples. In this paper, the geopolymer sample obtained is indicated as MK.

#### 2.2.2. Geopolymer Mortars

Table 2 shows the composition of the geopolymer-based mortars studied in the present paper. The samples were obtained by adding different percentages by weight of the ceramic wastes (in the range of 6–45 wt.%) to the freshly prepared geopolymer suspension, prepared as described in the previous paragraph, and quickly incorporating by controlled mechanical mixing (5 min at 800 rpm). On the other hand, the mass percentages of the ceramic waste were chosen in order to not change significantly the workability, setting times, and physical-mechanical properties. Ceramic waste was ground before use to obtain a fine powder (particle size in the range of 5–80 µm).

The mixture resulted well workable for several hours (the complete crosslinking and hardening took place in 5–7 h at room temperature, 20 °C). The mortar samples are hereafter indicated as PS-MK, where PS refers to pressed burnt and extruded ceramic waste; RP-MK, where RP refers to raw pressed ceramic waste; RP_dry_-MK, where RP_dry_ refers to raw pressed ceramic waste annealed at 750 °C for 5 h in air; Gy-MK, where Gy refers to gypsum waste; and MIX-MK, which refers to geopolymer mortars obtained from the addition of all ceramic wastes.

#### 2.2.3. Curing Treatments

As soon as prepared, all the specimens were cast in cubic moulds (50 × 50 × 50 mm^3^) and cured in >95% relative humidity conditions at 60 °C for 24 h. Subsequently, the specimens were kept at room temperature for a further 6 days in >95% relative humidity conditions, and then for a further 21 days in air, as reported in refs. [30,31,32,33,34,35,36,37,38].

### 2.3. Methods

SEM analysis was carried out using a Phenom Pro X Microscope (Phenom-World B.V., Eindhoven, The Netherlands) on fresh fracture surfaces, after metallization with gold, carried out using a high vacuum sputter-coating technique. The acceleration potential used was between 5 and 15 kV. The EDS analysis was conducted with a BSD detector in full mode.

Hydrostatic weighing for apparent density and open porosity measurements was carried out employing a balance OHAUS-PA213 provided by Pioneer.

To determine the water absorption of mortar specimens, 3 cubes from each series were oven-dried at a temperature of 60 °C for 24 h [39], and their weight was determined as starting weight. The samples were then immersed in water for 24 h and their saturated surface dry weight was recorded as the final weight. Water absorption of specimens was reported as the percentage increase in weight.

X-ray diffraction patterns were obtained with Ni-filtered Cu-K_α_ radiation (λ = 0.15406 nm) at room temperature (20 °C) with an automatic Rigaku powder diffractometer mod. Miniflex 600, operating in the θ/2θ Bragg-Brentano geometry. The phase recognition was carried out by using the PDF-4+ 2021 (International Centre for Diffraction Data^®^) database and the Rigaku PDXL2 software.

The compressive strength was evaluated according to EN 196-1 and measured by testing cubic concrete specimens (40 × 40 × 40 mm^3^) in a Controls MCC8 multipurpose testing machine with a capacity of 100 kN. The tests were performed after 28 days of curing at room temperature, and the values reported are the averages of 5 compression strength values.

The compression tests were performed until the sample was densified and/or ruptured at a constant displacement velocity of 2 kN/s.

## 3. Results and Discussion

### 3.1. X-ray Diffraction Characterization

The diffraction patterns of the raw materials used in this work, and the corresponding geopolymer samples and mortars are reported in Figure 1 and Figure 2, respectively. In Table 3, the corresponding degrees of crystallinity are reported.

As far as the raw materials, the porcelain stoneware waste (Figure 1A (PS)) consists of two main crystalline phases: quartz (SiO_2_), the main one, and sillimanite (Al_2_SiO_5_). Considering the range of composition of different PS products [40], this sample can be considered representative of this kind of material. An amorphous halo with a maximum at 2θ ≈ 23° is also present (the *x_c_* of the sample is ≈87%, Table 3), probably attributable to the melting of part of the alumina present in the system during the stoneware production [41,42]. This amorphous phase is responsible for the reduction of the characteristic porosity of ceramic materials fired at temperatures below 1000 °C, the characteristics of waterproofing properties of the stoneware, and resistance to acids and bases, and an increase in their mechanical properties compared to porous paste ceramics [41,42].

As far as the gypsum waste (Figure 1A (Gy)), it shows, apparently, just a single crystalline phase: gypsum. The diffraction pattern of the metakaolin sample (Figure 1A (metakaolin)) is characterized by a broad amorphous halo centred at 23° with only a crystalline peak at 25.4°, indicating the presence of small amounts of titanium oxide in the form of anatase (*x_c_* = 8%, Table 3).

RP and RP_dry_ waste (Figure 1B) present appreciably different diffraction patterns, although it is apparent that they are made of different amounts of the same crystalline phases: quartz, albite, and, in the case of the RP sample, kaolinite. The differences are attributable to the thermal treatment at 750 °C of the RP raw to prepare the RP_dry_ sample, which causes the transformation of kaolinite into metakaolin, a fact that is clearly pointed out by the disappearance in the diffraction pattern of the annealed sample at the diffraction peaks of 2θ = 12.5, 13.9, and 21 deg. that are distinctive of kaolinite crystalline structure, and the formation of an appreciable amorphous halo (the degree of crystallinity of the RP waste sample is 97%, while that of RP_dry_ waste is reduced to 78% upon annealing, Table 3) with a maximum intensity at 2θ ≈ 23°, in good agreement with the characterization of the diffraction pattern of MetaMax^®^ metakaolin reported in Figure 1A. As reported in Table 3, the estimated amount of glassy phase in these samples ranges from 92% in the case of metakaolin (as high as expected for this typically amorphous material), to 3% in the case of RP waste.

Finally, the presence of calcium compounds in the samples was excluded by EDX analysis (Table 4 reports the results in the case of PS and RP waste). This analysis was particularly important since, as reported in the literature [43], Ca ions negatively interfere with the geopolymerization reaction, giving the final material poor mechanical properties. In this way, as will be better described below, it was possible to create a geopolymer mortar by loading the mixture with an appreciable quantity of gypsum as filler, without significantly affecting the mechanical properties of the material obtained.

As far as the geopolymeric materials, the presence in the X-ray diffraction patterns (Figure 2) of an amorphous halo (characterized by a maximum centred at 2θ ≈ 29°) linked to the formation of a disordered network of Si–O–Al bonds shows that all the samples have undergone geopolymeric activation. This amorphous network is generated mainly by the reaction of metakaolin with sodium silicate. The amorphous hump is particularly evident for the geopolymer MK sample (Figure 2A (MK)) obtained from metakaolin only, without the addition of aggregates, in which the crystalline reflection at 2θ ≈ 25.4° is due to the presence in the starting metakaolin of a minor amount of TiO_2_. Similarly, in the diffraction patterns of all the other samples, crystalline reflections due to unreacted crystalline phases already present in the starting raw materials, or to new crystals formed during the geopolymerization reaction, are still present. It is worth noticing that the presence of these crystalline domains allows the activated material to limit the shrinkage phenomena due to the loss of water that occurs during the thermal curing and the ageing of the samples [43].

In particular, as far as the PS-MK geopolymer sample, the diffraction pattern (Figure 2A (PS-MK)) shows an amorphous halo more pronounced than that shown by the starting raw material (Figure 1A (PS)), thus suggesting that the geopolymerization reaction was successfully carried out (with reference to Figure 1 and Figure 2 and Table 3, the degree of crystallinity of the geopolymeric mortar is 65%, significantly lower than the 87% of the starting PS waste). It is not possible to observe direct evidence of the amorphous phase involvement of the porcelain stoneware in the geopolymeric reaction during basic activation. This is in line with literature finds [44], stating that crystalline phases do not take part in the geopolymerization reaction and do not significantly influence it.

Meanwhile, the system that shows major variations after the alkaline activation process (Figure 2A) is the Gy-MK. In fact, even if it is well known that gypsum is not involved in geopolymerization, in the used experimental conditions this crystalline phase is transformed into calcite (CaCO_3_) and thenardite (Na_2_SO_4_). The first phase is likely to be formed upon reaction, in the basic environment, of calcium ions with atmospheric carbon dioxide. Thenardite, on the other hand, is likely to be formed from the SO_4_^2−^ anions (deriving from gypsum) with the Na^+^ ions dissolved in the reagent base solution. As will be discussed in the next part, these ionic species greatly contribute to limiting the mechanical, physical, and chemical properties of the formed products, since, for example, the crystalline domain of calcite shows poor mechanical properties and low resistance to acids, bases, and water. Also, in the diffraction pattern of this sample, the presence of a broad hump centred at 2θ ≈ 29° is evident, pointing out the success of the geopolymerization.

If we compare the X-ray diffraction patterns of the RP-MK and RP_dry_-MK geopolymeric materials (Figure 2B (RP-MK) and (RP_dry_-MK)) with those of the starting raw materials (RP and RP_dry_ patterns of Figure 1B), the geopolymer mortars show the formation of a more pronounced amorphous halo, with a maximum shifted to 2θ ≈ 29°. In particular, the RP-MK sample shows a smaller decrease in the degree of crystallinity with respect to RP_dry_-MK (Δ*x_c_* = 38% in the case of RP-MK and Δ*x_c_* = 45% for RP_dry_-MK) if compared to the starting raw materials (*x_c_* = 60% and 43%, respectively, as reported in Table 3). This more pronounced reduction in the degree of crystallinity could be attributed to the fact that in the formation of the RP_dry_-MK mortar the geopolymerization reaction involved not only the added metakaolin (see Table 2 for the mix design), but also the metakaolin formed during the annealing at 750 °C of the RP sample, as discussed before.

Finally, Figure 2 shows the X-ray diffraction pattern of a geopolymer mortar (Figure 2, MIX-MK) obtained by using the mixture of all the treated waste reported in Table 2. The obtainment of this particular sample was derived from the necessity of valorizing the gypsum waste that, as discussed before in the text, usually has a strong detrimental effect on the properties of the final geopolymeric material since the presence of Ca ions inhibit the geopolymerization reaction [43]. As a matter of fact, the mix design of the MIX-MK sample reported in Table 2 has been tailored to incorporate in the geopolymer mortar the maximum quantity of gypsum without having a significant decrease in the mechanical properties of the new material.

### 3.2. Microstructural Analysis

To highlight the microstructure of the samples, SEM micrographs at different scale ranges are reported in Figure 3 for waste aggregates, while the SEM micrographs of freshly obtained fracture surfaces of the geopolymeric samples after setting are reported in Figure 4.

It is apparent that PS powders (see Figure 3A,B) appear as a collection of particles with irregular shapes and sharp edges, whose dimensions are in the range of 0.2–100 μm. RP and RP_dry_ aggregates (see Figure 3C–F) occur as particles of a few microns (0.5–5 μm) that are organized into larger aggregates. Finally, Gy powders appear as aggregates of elongated crystals of calcium sulfate with dimensions in the range of 1–10 μm.

The neat geopolymer sample (Figure 4A) shows a homogeneous amorphous structure with some cracks (that could have been produced when the sample was fractured to obtain a fresh fracture surface to be analyzed). The largely homogeneous microstructure suggests a good geopolymerization behaviour (as pointed out also by the X-ray diffraction analysis discussed in the previous paragraph). The morphology is very similar to that observed for the neat geopolymer sample and is observable also in the case of the geopolymeric mortar specimens (Figure 4B–F). In the case of the PS-MK sample (Figure 4B), the morphology is characterized by the presence of PS filler particles well-embedded into a homogeneous and compact geopolymer matrix and strongly adhering to it. No apparent fractures are present. Unlike what has been reported in the literature for analogous samples [10,11], the point examination of Figure 4B shows that the PS particles retain their irregular shape with rather sharp and well-defined edges. For this reason, it can be said that there are no clear indications of the possible involvement in the geopolymerization reaction of the PS particles, which therefore behave like a simple filler. As far as the morphology and microstructure of the RP-MK and RP_dry_-MK samples (Figure 4C,D, respectively), it is very similar and for both of them is not possible to recognize the waste aggregates. Also, in this case, the sample morphology is in line with the diffraction profile, where there is a major amorphous phase in RP_dry_-MK (the heat treatment at 750 °C for 5 h allowed the kaolinite calcination process that produced a major percentage of amorphous phase to RP-MK). It is worth pointing out that these mortars show a more consistent microstructure with respect to the geopolymer paste, with a reduced amount of microcracks.

A different morphology was observed in the case of the Gy-MK sample (Figure 4F), in which it is possible to identify the presence of crystals of calcium carbonate and sodium sulfate distributed in the geopolymer matrix. Finally, the MIX-MK sample (Figure 4E) also shows a morphology rather homogeneous, where, as in the case of the PS-MK sample, it is possible to recognize aggregate particles of PS with dimensions of about 100 μm strongly interpenetrated in the geopolymer matrix. Concluding, in all cases this homogeneous structure strongly suggests that ceramic waste does not take part, interfere, or inhibit the geopolymerization reaction.

### 3.3. Physical and Mechanical Properties

Table 5 reports the compressive strength, density, and water absorption values of the geopolymeric samples prepared.

By comparing the samples’ densities values (first row of Table 5) it is apparent that, as expected, all the geopolymeric mortars are characterized by values fairly higher than the neat geopolymer. This is due to the presence of a significant amount (see Table 2 for the mix design) of the aggregate that in every case is made of dense materials (the typical density of porcelain stoneware and pressed clays is about 2.6 g/cm^3^).

The only exception is the Gy-MK mortar, in which the density is practically the same as the neat geopolymer. This fact is in line with the high water absorption capability of the sample, due to the high affinity of the calcium sulphate with water. This fact decreases the workability of the geopolymer slurry, strongly reducing the set time and thus causing the incorporation of a greater amount of air than the other mortars, resulting in a final product with the lowest density value (1335 kg/m^3^).

A similar trend can be pointed out by examining the values of the compressive strength of the different samples (third line of Table 5). As we can see, except for the Gy-MK mortar, all the other samples show a comparable or even higher compressive strength to the neat geopolymer, with an increase of compressive strength values up to 40%, indicating that the addition of the aggregates has a remarkable enhancing effect on the mechanical properties of the materials. This is likely to be caused by the good adhesion and the good dispersions of the filler in the geopolymer matrix, as shown by the SEM images reported in Figure 4, which suggests that the waste particles dispersed within the geopolymer matrix can create a barrier against crack growth, enhancing the mechanical response of the material.

It is worth pointing out that the most evident improvement in the mechanical properties of the material has been recorded for the RP_dry_-MK sample. This important increase of the compressive strength is probably due to the presence in the starting sample of a greater quantity of reactive phase (i.e., the metakaolin added according to the mix design of Table 2 and produced by the calcination of kaolinite at 750 °C). In this way, as also highlighted by the SEM images shown in Figure 2, the geopolymerization reaction led to the obtaining of a geopolymer mortar sample in which the non-reactive crystalline phase is very well-included and dispersed in the geopolymer amorphous matrix.

At variance with this sample, the Gy-MK mortar shows very poor mechanical properties, with a reduction of about 85% in the compressive strength value of the neat geopolymer (MK). Substantially, the presence of gypsum within the paste negatively affects the mechanical properties of the final product because gypsum is a source of Ca ions, which are competitive with the geopolymerization reaction [22,43]. As already discussed, to find a possible strategy to valorize this very abundant waste, an alternative sample composition was developed, in which gypsum was mixed (which hurts the mechanical properties of the final material) with the other wastes, and in particular, RPdry, which significantly improves the mechanical properties of the mortars (as seen in the previous sections). The mix obtained (see Table 2) allowed to include up to ≈7% by weight of gypsum without significantly compromising the mechanical properties of the final product, a result that to our knowledge was never previously obtained using aggregates with high reactive calcium concentration. As a final note, it is worth highlighting that the material obtained presents good mechanical properties despite its high water absorption capacity (>25 wt.%), which is fairly higher than that of the neat geopolymer (18 wt.%).

### 3.4. Applications in the Field of the Creative Industry and Cultural Heritage

A preliminary investigation of the potentialities of the geopolymer-based materials made from ceramic wastes described so far has shown that such systems could be used for the restoration and conservation of artworks and the creation of products for art and design.

Experimental tests have shown that the geopolymer slurry is characterized by very good thixotropic behaviour and high workability, which makes it easy to spread and model on different substrates that need repair interventions. Moreover, the chemical compatibility of the geopolymer-based materials with different substrates (tuff, cement, ceramic, and porcelain stoneware) ensures good adhesion with them, pointing to the possibility of using this kind of binder as a joining or fixing paste.

To this aim, Figure 5 reports some artefacts made of porcelain stoneware (top of Figure 5) and ceramic (bottom of Figure 5) that were restored and repaired by using the previously described PS-MK and RP_dry_-MK geopolymer-based pastes as fixing materials. Particularly, the two objects were subsequently damaged and then repaired using geopolymer systems as bonding materials. The repair process remained stable even after many months. A very good adhesion in each of the examined cases was observed, even for those materials containing an elevated concentration of filler with respect to others described in the literature [13]. This encouraging result was confirmed by performing SEM characterization of the interface between the geopolymer mortar, used as repairing material, and a pottery fragment, as shown in Figure 6: it is also apparent that at a micrometric level, a very strict adhesion between the geopolymer and ceramic phases can be observed, and, also at micrometric level, the geopolymer material and the ceramic substrate seem to form a continuous phase with no neat interface and no fractures. However, physical properties with a focus on the optimization of adhesion between ceramic artifacts and geopolymer binders should be further studied. For this reason, the incorporation of organic additives in geopolymer formulations developed for this aim is still in progress.

These preliminary results suggest that the proposed mortars can be applied as materials to decorate, seal, and repair cracks and fractures in stone, artifacts, tiles, and masonries.

To perform these functions optimally, they must have aesthetic characteristics similar to those of the materials to be repaired, but, as suggested by current restoration practice, still well recognizable: as shown in the lower part of Figure 5, mortars of the same colour as the artifact, but slightly paler, were obtained.

Finally, it is worth noting that the geopolymer products developed and characterized in this work can be easily coloured by simply adding water-based pigments into the slurry and/or through post-panting operations with oil-based paint and/or cold painting. As an example, Figure 7 reports different kinds of artefacts realized with PS-MK mortar, with the addition of water-based pigments. Moreover, the colour of the mortar can be changed by adding rock powders, mimicking the material to be restored.

Future developments will be focused on extending the methodology developed for obtaining building elements with high performance in terms of mechanical strength (rigidity of shape, texture, and geometry), technological integrability (with other materials), low maintenance, substitutability, recoverability at end of life, thermal and acoustic insulation, and architectural and aesthetic characteristics. Moreover, research will be integrated with Life Cycle Assessment (LCA) studies and a deep investigation of the social and economic dimensions, using Life Cycle Costs (LCC) and Social Life Cycle Assessment (S-LCA).

## 4. Conclusions

Geopolymers are environmental-friendly materials for which their use in many application fields has been proposed. To date, a great effort has been made as far as the structural applications and the replacement of ordinary Portland cement with this class of materials are concerned. However, in the last years, their application in art and design, and more in general, in Cultural Heritage is attracting more and more attention. As a matter of fact, geopolymer-based materials may find interesting applications in these fields thanks to their interesting chemical and physical properties (such as high durability and good flexural and compressive strength). Moreover, these properties can be also tailored in order to guarantee functional and aesthetic compatibility with the original materials on which the restoration action has to be performed.

The present research describes the preparation and characterization of new geopolymer mortars obtained by recycling wastes deriving from the production process and the “end of life” of porcelain stoneware products. In particular, geopolymer mortars were obtained after the consolidation of a slurry prepared by using metakaolin, an activating alkaline solution, and different kinds of by-products of tiles and ceramic production (pressed burnt and extruded ceramic waste, raw pressed and gypsum resulting from exhausted moulds). The obtained materials resulted to be largely composed of an amorphous binding material in which the filler was homogeneously dispersed.

The chemical compatibility of the geopolymer materials with different substrates (tuff, cement, ceramic, and porcelain stoneware) ensures good adhesion with them, pointing to the possibility of using this kind of binder as a joining and fixing material or as sacrificial material for the restoration of stone objects. To this aim, the possible addition of rock powders within the mix design composition could allow obtaining materials that mimic different types of stone, thus reaching good aesthetic compatibility. In particular, ongoing studies on formulations including volcanic rocks and tuffs show promising perspectives in terms of chemical, mineralogical, mechanical, and aesthetic compatibility with the local built heritage since they mimic the traditional materials used in the Vesuvius archaeological area. In this framework, to further increase the adhesion between the substrate and geopolymer binders, the incorporation of organic additives in geopolymer formulations developed for this aim is also in progress. The possibility to develop new materials with the concurrent reduction of their environmental footprint and production cost can create a tangible gain in the sectors of the decorative industry and Cultural Heritage. In fact, the demonstration of real and practical use of sustainable materials for the creative industry and restoration field can improve business around waste collection and conversion. Moreover, continued use of recycled materials in mass production can increase the competitive position of Eco-materials on the market and reduce the overall environmental footprint in the creative industry and Cultural Heritage field.

In this scenario, the authors wish to give a contribution to the research field of sustainable materials to enhance the use of recycled raw materials and also in the field of Art and Design, restoration, conservation, and/or rehabilitation of historic monuments, decorative and architectural intervention.

## Figures and Tables

**Figure 1 materials-15-08600-f001:**
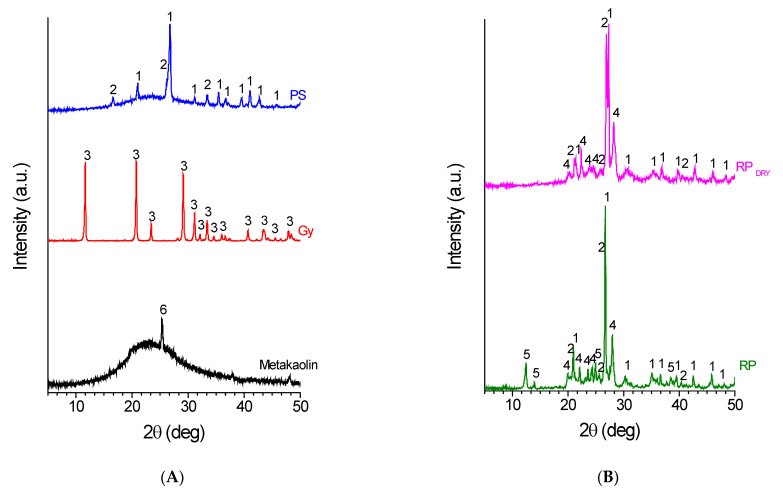
X-ray diffraction pattern of raw materials. Part (**A**): porcelain stoneware (PS), gypsum (Gy), and metakaolin; Part (**B**): Raw pressed (RP) and annealed raw pressed (RP_dry_) waste. The numbers correspond to main diffraction peaks (with relative ICDD PDF-4+ 2021 card numbers) of 1-Quartz (SiO_2_) (01-083-0539); 2-Sillimanite (Al_2_SiO_5_) (01-088-0893); 3-Gypsum (CaSO_4_·2(H_2_O)) (00-003-0053); 4-Albite (NaAlSi_3_O_8_) (01-089-6427); 5-Kaolinite (Al_2_Si_2_O_5_(OH)_4_) (01-080-0886); and 6-Anatase (TiO_2_) (01-070-7348).

**Figure 2 materials-15-08600-f002:**
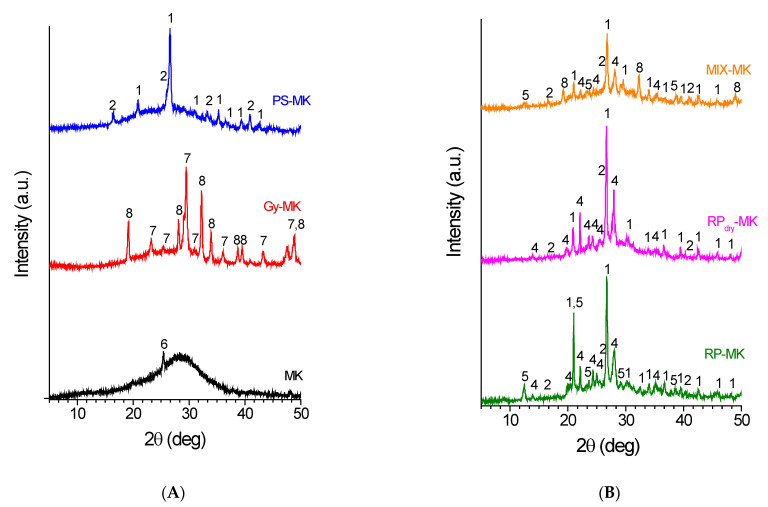
X-ray diffraction pattern of (**A**) geopolymer (MK) and geopolymeric mortars loaded with porcelain stoneware waste (PS-MK), gypsum waste (Gy-MK), and (**B**) geopolymeric mortars loaded with raw pressed (RP-MK) and annealed raw pressed (RP_dry_-MK) waste. Also in (**B**), the X-ray diffraction pattern of the MIX-MK mortar obtained with the aggregate mixtures described in Table 2 is reported. The numbers correspond to the main diffraction peaks of 1-Quartz (SiO_2_) (01-083-0539); 2-Sillimanite (Al_2_SiO_5_) (01-088-0893); 3-Gypsum (CaSO_4_·2(H_2_O)) (00-003-0053); 4-Albite (NaAlSi_3_O_8_) (01-089-6427); 5-Kaolinite (Al_2_Si_2_O_5_(OH)_4_) (01-080-0886); 6-Anatase (TiO_2_) (01-070-7348); 7-Calcite (CaCO_3_) (01-080-9776); and 8-Thenardite (Na_2_SO_4_) (04-010-2457).

**Figure 3 materials-15-08600-f003:**
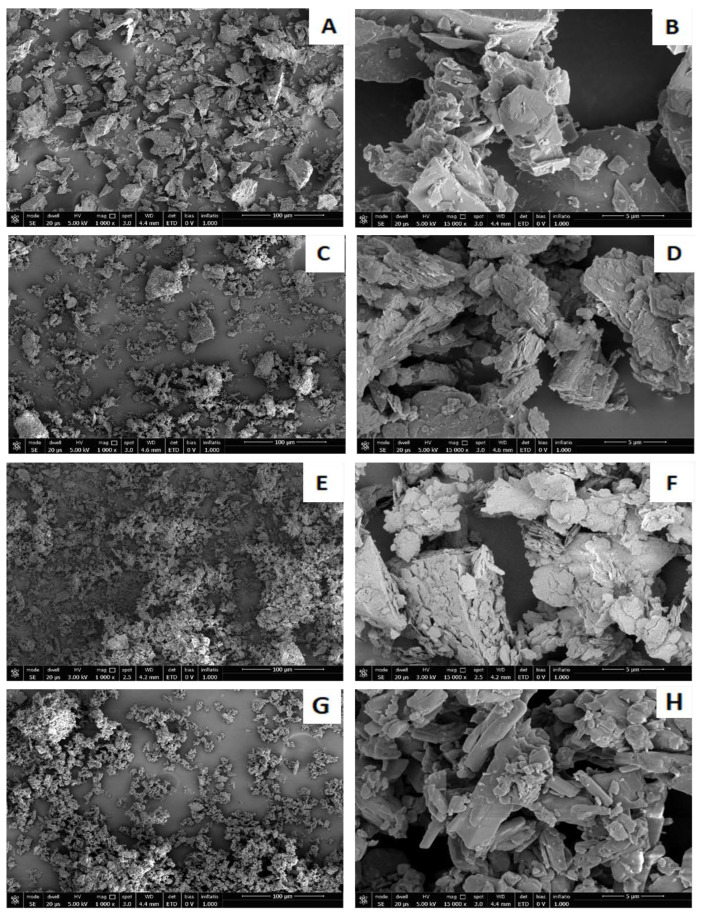
Scanning Electron Microscope (SEM) images of PS (**A**,**B**), RP (**C**,**D**), RP_dry_ (**E**,**F**), and Gy (**G**,**H**) waste at 1000× and 15,000× magnifications.

**Figure 4 materials-15-08600-f004:**
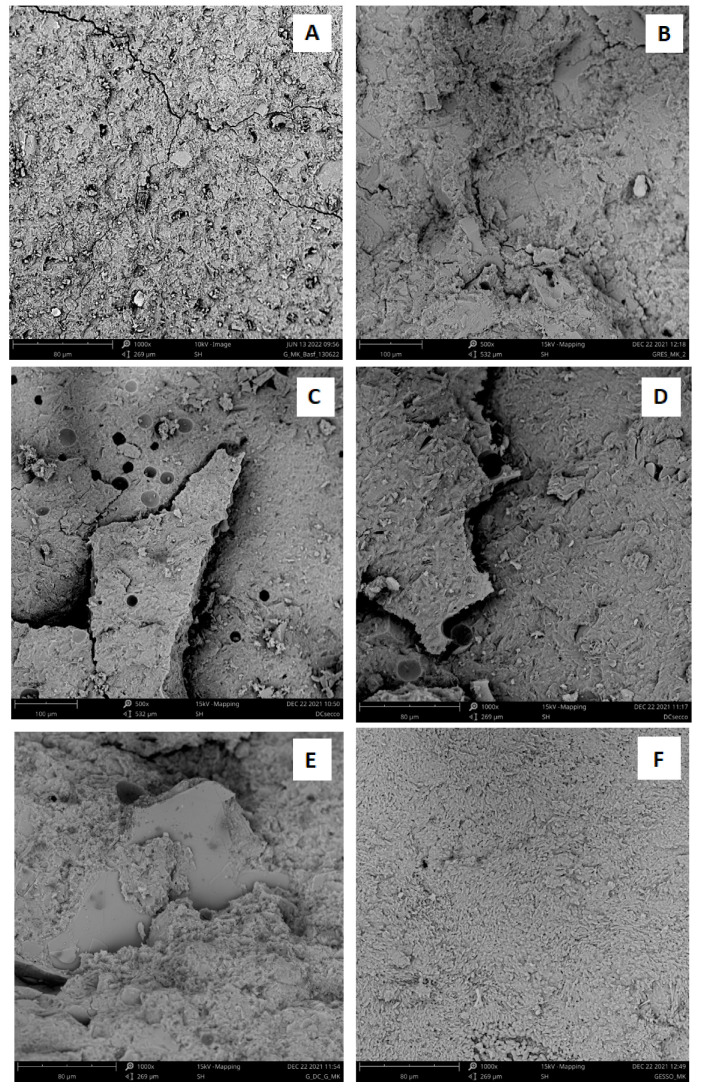
Scanning Electron Microscope (SEM) images of (**A**) neat geopolymer; (**B**) PS-MK; (**C**) RP-MK; (**D**) RP_dry_-MK; (**E**) MIX-MK; and (**F**) Gy-MK mortars at 1000× magnifications.

**Figure 5 materials-15-08600-f005:**
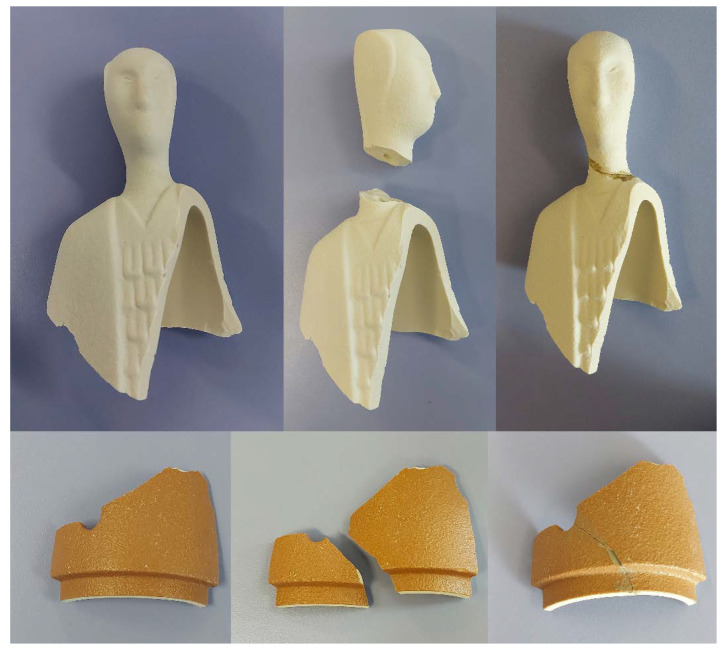
Porcelain stoneware and pottery artworks restored and repaired by using the geopolymer-based materials PS-MK and RP_dry_-MK described in this paper.

**Figure 6 materials-15-08600-f006:**
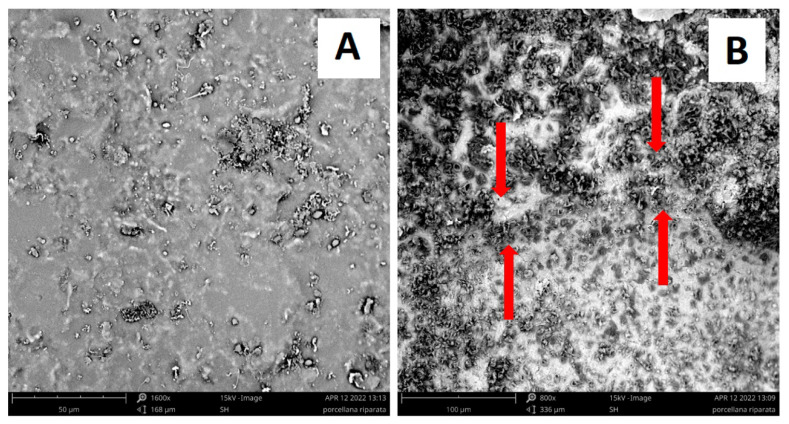
Scanning electron microscope (SEM) micrographs of (**A**) ceramic substrate and (**B**) interface transition zone (indicated by the red arrows) between the ceramic substrate (upper part of the figure) and RP_dry_-MK geopolymer mortar (low part of the figure).

**Figure 7 materials-15-08600-f007:**
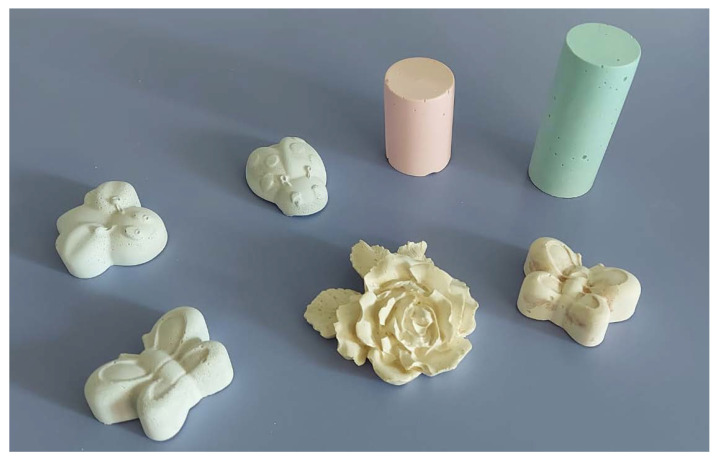
Examples of coloured artefacts created with the geopolymer-based materials described in this paper. As apparent, even objects with complex shapes can be easily realized by a simple casting procedure.

**Table 1 materials-15-08600-t001:** Chemical composition (weight %) of the metakaolin BASF MetaMax^®^ and sodium silicate solution used in this paper.

Compound	Metakaolin	Sodium Silicate
SiO_2_	52.2	27.40
Al_2_O_3_	45.1	-
Na_2_O	0.22	8.15
K_2_O	0.15	-
TiO_2_	1.75	-
Fe_2_O_3_	0.42	-
CaO	0.04	-
MgO	0.04	-
P_2_O_5_	0.08	-
H_2_O	-	64.45

**Table 2 materials-15-08600-t002:** Mix design of the geopolymeric samples prepared in this work (MK: neat geopolymer; Gy: gypsum waste; RP: raw pressed ceramic waste; PS: porcelain stoneware waste). In the MIX-MK sample, a mixture of all the used waste was used.

Materials (wt.%)	MK	PS-MK	RP-MK	RP_dry_-MK ^(1)^	Gy-MK	MIX-MK
Metakaolin	37.5	11	11.8	11.6	15.3	11
NaOH	7.2	5.2	5.6	5.6	7.3	5.2
Sodium silicate	55.3	40	43	42.7	56.1	40
Pressed burnt and extruded ceramic waste	-	43.8	-	-	-	20.9
Raw pressed	-	-	39.6	40.1	-	16.3
Gypsum	-	-	-	-	21.3	6.6

^(1)^ The raw pressed clay was dried in an oven at 750 °C for 5 h before use.

**Table 3 materials-15-08600-t003:** Degrees of crystallinity (*x_c_*, %) of the starting raw waste and corresponding geopolymeric materials obtained by geopolymerization reaction, whose diffraction patterns are reported in Figure 1 and Figure 2.

Sample Waste	*x_c_* (%)	Geopolymeric Sample Loaded with	*x_c_* (%)
metakaolin	8	geopolymer (MK)	7
porcelain stoneware (PS)	87	porcelain stoneware (PS)	65
gypsum (Gy)	90	gypsum (Gy)	61
raw pressed waste (RP)	97	raw pressed waste (RP)	60
annealed raw pressed waste (RP_dry_)	78	annealed raw pressed waste (RP_dry_)	43
--		MIX waste	42

**Table 4 materials-15-08600-t004:** Chemical composition (weight %) of PS and RP used in this paper as obtained by EXD characterization.

Phase	PS	RP
SiO_2_	64.3	65.4
Al_2_O_3_	29.1	29.1
Na_2_O	2.45	2.38
K_2_O	3.51	2.19
Other	0.64	0.93

**Table 5 materials-15-08600-t005:** Physical and mechanical properties of the geopolymer sample (MK) and geopolymeric mortars obtained with porcelain stoneware waste (PS-MK), raw pressed waste (RP-MK), calcined raw pressed waste (RP_dry_-MK), and gypsum (Gy-MK) as filler, and of the geopolymeric mortar with the mixed filler whose composition is reported in Table 2 (MIX-MK). The calcined raw pressed clay (RP_dry_) was obtained by annealing as obtained raw pressed clay (RP) in an oven at 750 °C for 5 h.

Sample Properties	MK	PS-MK	RP-MK	RP_dry_-MK	Gy-MK	MIX-MK
Density (kg/m^3^)	1370	1773 ± 95	1718 ± 80	1687 ± 90	1335 ± 99	1627 ± 77
Water absorption (%)	18 ± 1	16 ± 1	18 ± 1	19 ± 1	>25	>25
Compressive strength (MPa)	25 ± 2	30 ± 1	25 ± 3	41 ± 3	4.0 ± 0.5	38 ± 2

## Data Availability

Not applicable.

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
