# Peer review of "Development of Geopolymer-Based Materials with Ceramic Waste for Artistic and Restoration Applications"

_materials, 2022, doi:10.3390/ma15238600_

Round 1

Reviewer 1 Report

The manuscript requires major revision.

Below I list my comments and questions.

Abstract, Table 2. Is it correct that the name of the material is “Raw pressed”?

P. 2, line 48. “Lead, Fluorine, Boron”. Please write the names of chemical elements with small letters.

The sum of all components of metakaolin listed Table 1 gives 100,19% instead of 100%. Please correct the data listed in the table.

In the sample preparation section, the composition of the solution is expressed as Na2O 1.55 SiO2 12.14 H2O, while the composition of the whole geopolymer system is Al2O3 3.48 SiO2 1.0 Na2O 12.14 H2O. Please explain the reasons for selection of these compositions or provide references.

Line 146. “…the composition of the whole geopolymer system can be expressed as Al2O3 3.48 SiO2 1.0 Na2O 12.14 H2O, assuming a complete geopolymerization process.” Please explain the meaning of the term “a complete geopolymerization process”.

Table 2. Please explain how you choose the ratio of the components in the geopolymeric samples.

Line 159. “…RC-MK where RC refers to raw pressed ceramic waste …” In Table 2, there is a different notation: “RP: raw pressed ceramic waste”. Please make correction.

Line 201, Table 3. “…the corresponding degrees of crystallinity are reported”. Please explain how you determine the degree of crystallinity.

Fig. 1A. Please discuss the XRD pattern of metakaolin. Why its crystallinity fraction is so low? Why there are no peaks of components constituting this material. There are only the peaks attributed to anatase, while the content of TiO2 in metakaolin is only 1.74 wt%.

Please provide the standard XRD cards numbers for all the crystalline phases that you identified.

It seems that the position of the main peak on the XRD pattern of metakaolin does not match that of anatase, which is at 2theta = 25.3 deg.

In Figs 1 and 2, there are many peaks of sillimanite that are not marked with number 2. In the standard card of sillimanite, there are two main peaks located at 2theta = 25.9 deg. and 26.4 deg. with intensities of 81% and 100%, respectively. Such peaks are clearly seen in XRD patterns of PS and PRdry presented in Fig. 1 and in the XRD pattern of PS-MK? Fig. 2. The positions of peaks marked with number 1 are different in different XRD patterns of the materials probably because they are superimposed on the peaks of silimanite. In the XRD pattern of the RP materials, please mark all peaks denoted by the number 1.

It seems that the peaks assigned to kaolinite do not exactly match the kaolinite standard XRD card.

Please compare the XRD patterns of RP-MK and Mix-RK in Fig. 2B. The peaks marked with number 5 are different. Please give your comments on this matter.

Line 239. “…the diffraction peaks at 2θ= 12.5, 13.9 and 21 deg, that are distinctive of kaolinite crystalline structure…” In the standard XRD cards of kaolinite, I cannot find the peak at 13.9 deg.

Line 245. “As a final note, it is worth pointing out that the chemical compositions of the ceramic waste used in this paper (porcelain stoneware PS, raw pressed RP), have been also checked by EDS analysis (see Table 4). The data are in good agreement with those obtained by X-ray diffraction characterization.” 1. The sum of all the components of PS and of RP is less than 100%. Please give your comments. 2. According to Table 4, both materials contain K2O. In XRD patterns of these materials there is no crystalline phase containing K2O. According to Table 4, PS material contains also Na2O. In XRD patterns of this material there is no crystalline phase containing Na2O. According to Table 4, PS material contains also Na2O. In XRD patterns of these materials there is no crystalline phase containing K2O. Please explain the meaning of the sentence “The data are in good agreement with those obtained by X-ray diffraction characterization.”

Please explain in more detail what is depicted in Fig. 5. There are two different images. What artwork was restored and repaired by using the geopolymer - based material PS-MK and what artwork was repaired using the RPdry-MK material?

Author Response

Dear Reviewer,

we would like to thank you for the suggestions and comments that have allowed us to improve significantly and clarify some of the key topics addressed in the paper.

In the following we report the list of changes that we have done to the manuscript in order to address all the reviewers' comments. Please note that all changes have highlighted in the revised text by using the “track-change” mode.

  1. Abstract, Table 2. Is it correct that the name of the material is “Raw pressed”?

Authors' answer:

In the ceramic industry, “raw pressed” is defined as the clay that is used to make ceramic material artifacts before firing, which is expelled in pressed form during the production process. Accordingly, no changes have been done on the manuscript.

  1. P. 2, line 48. “Lead, Fluorine, Boron”. Please write the names of chemical elements with small letters.

Authors' answer:

We thank the Reviewer for his/her comment. The authors have corrected the names of chemical elements according to the Reviewer’s suggestion.

  1. The sum of all components of metakaolin listed Table 1 gives 100,19% instead of 100%. Please correct the data listed in the table.

Authors' answer:

We thank the Reviewer for his/her comment. The authors have corrected the data according to the Reviewer’s suggestion.

  1. In the sample preparation section, the composition of the solution is expressed as Na2O 1.55 SiO2 12.14 H2O, while the composition of the whole geopolymer system is Al2O3 3.48 SiO2 1.0 Na2O 12.14 H2O. Please explain the reasons for selection of these compositions or provide references.

Authors' answer:

The authors have chosen to use the specific stoichiometries of alkaline activating solution and geopolymer based on their previous work (references 25-28 of the manuscript). In fact, based on the initial chemical composition of the metakaolin, the final composition of the geopolymer is the result of the best compromise between processability, rheological, chemical, and mechanical properties (reference 14 and 15 of the manuscript for example). Accordingly, no changes have been done on the manuscript.

  1. Line 146. “…the composition of the whole geopolymer system can be expressed as Al2O3 3.48 SiO2 1.0 Na2O 12.14 H2O, assuming a complete geopolymerization process.” Please explain the meaning of the term “a complete geopolymerization process”.

Authors' answer:

We thank the Reviewer for his/her comment. The authors have done some modifications to improve the text comprehension (lines 158-167 in Experimental Section).

  1. Table 2. Please explain how you choose the ratio of the components in the geopolymeric samples.

Authors' answer:

We thank the Reviewer for his/her comment. Also in this case the amount of waste added to the geopolymer mixture has been dosed and tested to achieve excellent workability, right setting times and at the same time good physical-mechanical properties. The text has been modified specifying these criteria.

  1. Line 159. “…RC-MK where RC refers to raw pressed ceramic waste …” In Table 2, there is a different notation: “RP: raw pressed ceramic waste”. Please make correction.

Authors' answer:

We thank the Reviewer for his/her comment. The mistakes have been corrected according to the Reviewer’s suggestion.

  1. Line 201, Table 3. “…the corresponding degrees of crystallinity are reported”. Please explain how you determine the degree of crystallinity.

Authors' answer:

The degree of crystallinity of a sample is defined as the fraction of the sample which is crystalline. There are many ways from which the degree of crystallinity can be determined, including X-ray analysis. In these cases, an X-ray diffraction pattern is arbitrarily divided into two parts (phases) under the assumption that one of them is crystalline and the other is amorphous. These areas are considered to be proportional to the phase volumes. In these assumptions the degree crystallinity is determined by calculating the ratio between the area subtended by crystalline peaks and that of the whole pattern (crystalline peak and amorphous halo). Since this is a very common way of expressing this quantity, no changes have been done to the text.

  1. Fig. 1A. Please discuss the XRD pattern of metakaolin. Why its crystallinity fraction is so low? Why there are no peaks of components constituting this material. There are only the peaks attributed to anatase, while the content of TiO2 in metakaolin is only 1.74 wt%.

Authors' answer:

We thank the Reviewer for his/her observation. In order to reply to this comment, we would like to point out that metakaolin consists mainly of an amorphous structure of silicon and aluminum oxides. This type of phase is widely visible in the amorphous halo with a center at 2θ = 23 degs. This is due to the heat treatment used to obtain the material from the starting kaolin. It is therefore not surprising that the only crystalline phase present is the anatase shown in the figure. Accordingly, no changes have been done on the manuscript.

  1. Please provide the standard XRD cards numbers for all the crystalline phases that you identified.

Authors' answer:

We thank the Reviewer for his/her observation. In order to satisfy this issue, all the card numbers are now indicated in the text.

  1. It seems that the position of the main peak on the XRD pattern of metakaolin does not match that of anatase, which is at 2theta = 25.3 deg.

Authors' answer:

We thank the Reviewer for his/her very careful reading of the manuscript. In order to satisfy this issue, we have specified in the text the XRD card number 01-070-7348 for anatase that is in perfect agreement with the experimental diffraction peak reported.

  1. In Figs 1 and 2, there are many peaks of sillimanite that are not marked with number 2. In the standard card of sillimanite, there are two main peaks located at 2theta = 25.9 deg. And 26.4 deg. with intensities of 81% and 100%, respectively. Such peaks are clearly seen in XRD patterns of PS and PRdry presented in Fig. 1 and in the XRD pattern of PS-MK? Fig. 2. The positions of peaks marked with number 1 are different in different XRD patterns of the materials probably because they are superimposed on the peaks of silimanite. In the XRD pattern of the RP materials, please mark all peaks denoted by the number 1.

Authors' answer:

We thank the Reviewer for his/her observation. Actually, we have not marked only very minor peaks just for the sake of readability of the graph.

  1. It seems that the peaks assigned to kaolinite do not exactly match the kaolinite standard XRD card.

Authors' answer:

We thank the Reviewer for his/her observation. In order to satisfy this issue, we have specified in the text the XRD card number 01-080-0886 that is in perfect agreement with the experimental diffraction peak reported.

  1. Please compare the XRD patterns of RP-MK and Mix-RK in Fig. 2B. The peaks marked with number 5 are different. Please give your comments on this matter.

Authors' answer:

We thank the Reviewer for his/her observation. Actually, as done in figure 1 and 2, for the sake of the clarity not all the diffraction peaks have been reported, but only the most intense ones.

  1. Line 239. “…the diffraction peaks at 2θ= 12.5, 13.9 and 21 deg, that are distinctive of kaolinite crystalline structure…” In the standard XRD cards of kaolinite, I cannot find the peak at 13.9 deg.

Authors' answer:

We thank the Reviewer for his/her observation. In order to comply with this issue, we have added the number of the XRD card phase (01-080-0886) that has been used for the identification of the phase and that matches all the experimental pattern.

  1. Line 245. “As a final note, it is worth pointing out that the chemical compositions of the ceramic waste used in this paper (porcelain stoneware PS, raw pressed RP), have been also checked by EDS analysis (see Table 4). The data are in good agreement with those obtained by X-ray diffraction characterization.” 1. The sum of all the components of PS and of RP is less than 100%. Please give your comments. 2. According to Table 4, both materials contain K2O. In XRD patterns of these materials there is no crystalline phase containing K2O. According to Table 4, PS material contains also Na2O. In XRD patterns of this material there is no crystalline phase containing Na2O. According to Table 4, PS material contains also Na2O. In XRD patterns of these materials there is no crystalline phase containing K2O. Please explain the meaning of the sentence “The data are in good agreement with those obtained by X-ray diffraction characterization.”

Authors' answer:

We thank the Reviewer for his/her observation. In order to reply to this issue, we would like to point out that EDS analysis does not allow obtaining the amount of the elements in their oxidation state or in their crystalline organization, so it is a common practice to express the content of the detected elements as the most common relative oxides at room temperature of the specific specie. Therefore, the reviewer should not be surprised by the lack of correspondence between the proposed phases and the oxides expressed in the table. When the authors claim that the EDX analysis confirms the crystalline phases found, it is meant that the chemical elements found are in such abundant ratios to justify the various crystalline phases presence. Accordingly, no changes have been done on the manuscript.

  1. Please explain in more detail what is depicted in Fig. 5. There are two different images. What artwork was restored and repaired by using the geopolymer - based material PS-MK and what artwork was repaired using the RPdry-MK material?

Authors' answer:

The authors have added a description of Figure 5 (lines 287-292 in the Result and Discussion section; file in review mode-all comments). Particularly, the best formulations were used (PS-MK and RPdry-MK materials) to bond and restore ceramic artefacts after breakage. For simplicity, we have shown pictures of porcelain stoneware and ceramic products restored once but this type of intervention, with the same results, has been performed with both mortars.

Reviewer 2 Report

The paper "Sustainable design of geopolymer-based materials for artistic and restoration applications" presents a relevant theme and within the scope of this journal, and can be considered after some corrections suggested below:

(a) The abstract is generally well written, however in terms of content it is generic, i.e., the authors lack an in-depth study of the quantitative results of this research;

(b) Scientific innovation is limited in the introduction of the paper, the authors must go deeper and detail what this research differs from countless others that exist on this topic, this must be evidenced together with the objectives at the end of the introduction;

(c) The state of the art of the evaluated topic needs to be improved by the authors, note that some topics are absent and need to be known with current research, such as: 10.1016/j.cscm.2017.03.001; 10.1016/j.cscm.2022.e01307; 10.1016/j.cscm.2021.e00802

(d) “It is well known that the geopolymerization reaction is based on the alkaline activation of an aluminosilicate raw material using a strongly alkaline solution. The latter was prepared by dissolving solid sodium hydroxide into the sodium silicate solution. The obtained solution was then allowed to equilibrate and cool for 24 hours. The composition of the overall solution can be expressed as Na2O 1.55 SiO2 12.14 H2O. Metakaolin was then incorporated into the activating solution with a liquid to solid ratio of 1.4:1 by weight and mixed by a mechanical mixer for 10 min at 800 rpm. As revealed by EDS analysis carried out on the cured samples, the composition of the whole geopolymer system can be expressed as Al2O3 3.48 SiO2 1.0 Na2O 12.14 H2O, assuming a complete geopolymerization process. In this paper, the geopolymer sample obtained is indicated as MK” Authors should better explain this passage and the origins of the raw material.

(e) “The neat geopolymer sample (Figure 4A) shows a homogeneous amorphous structure with some cracks likely die to the production of the fracture surface. A similar morphology is observable also in the case of the geopolymeric mortar specimens (Figures 3B– 330 F): in the case of the PS-MK sample (Figure 4B), its morphology is characterized by the presence of PS particles well embedded into the geopolymer matrix and strongly adhering to it.” Detail and explain this passage better.

(f) A topic on future perspectives should be inserted by the authors, considering the gaps found in this research.

Author Response

Dear Reviewer,

we would like to thank you for the suggestions and comments that have allowed us to improve significantly and clarify some of the key topics addressed in the paper.

In the following we report the list of changes that we have done to the manuscript in order to address all the reviewers' comments. Please note that all changes have highlighted in the revised text by using the “track-change” mode.

************** Reviewer 1

The paper "Sustainable design of geopolymer-based materials for artistic and restoration applications" presents a relevant theme and within the scope of this journal, and can be considered after some corrections suggested below:

(a) The abstract is generally well written, however in terms of content it is generic, i.e., the authors lack an in-depth study of the quantitative results of this research;

Authors' answer:

We thank the Reviewer for his/her observation. The abstract has been modified according to the Reviewer’s suggestion.

(b) Scientific innovation is limited in the introduction of the paper, the authors must go deeper and detail what this research differs from countless others that exist on this topic, this must be evidenced together with the objectives at the end of the introduction;

Authors' answer:

The introduction has been modified by adding information and considerations about the innovation and differences between the paper and other research papers on this topic (lines 69-77 and lines 104-120). Accordingly, some new bibliographical references (9-14) have been added

(c) The state of the art of the evaluated topic needs to be improved by the authors, note that some topics are absent and need to be known with current research, such as: 10.1016/j.cscm.2017.03.001; 10.1016/j.cscm.2022.e01307; 10.1016/j.cscm.2021.e00802

Authors' answer:

The state of the art has improved by adding the bibliographical references indicated by the Reviewer.

(d) “It is well known that the geopolymerization reaction is based on the alkaline activation of an aluminosilicate raw material using a strongly alkaline solution. The latter was prepared by dissolving solid sodium hydroxide into the sodium silicate solution. The obtained solution was then allowed to equilibrate and cool for 24 hours. The composition of the overall solution can be expressed as Na2O 1.55 SiO2 12.14 H2O. Metakaolin was then incorporated into the activating solution with a liquid to solid ratio of 1.4:1 by weight and mixed by a mechanical mixer for 10 min at 800 rpm. As revealed by EDS analysis carried out on the cured samples, the composition of the whole geopolymer system can be expressed as Al2O3 3.48 SiO2 1.0 Na2O 12.14 H2O, assuming a complete geopolymerization process. In this paper, the geopolymer sample obtained is indicated as MK” Authors should better explain this passage and the origins of the raw material.

Authors' answer:

We thank the Reviewer for his/her comment. Lines 145-148 and 158-167 in the Experimental Section have been modified for the sake of the clarity and to improve better explain the origins of the raw materials.

(e) “The neat geopolymer sample (Figure 4A) shows a homogeneous amorphous structure with some cracks likely die to the production of the fracture surface. A similar morphology is observable also in the case of the geopolymeric mortar specimens (Figures 3B– 330 F): in the case of the PS-MK sample (Figure 4B), its morphology is characterized by the presence of PS particles well embedded into the geopolymer matrix and strongly adhering to it.” Detail and explain this passage better.

Authors' answer:

Lines 188-201 and 214-16 in the Result and Discussion section have been modified to add details and clairy the passage as requested.

(f) A topic on future perspectives should be inserted by the authors, considering the gaps found in this research.

Authors' answer:

Several considerations on the future perspective of the research presented in the paper have been added in the Conclusion section according to the Reviewer’s suggestion.

Reviewer 3 Report

REVIEW

on article

 Sustainable design of geopolymer-based materials for artistic

and restoration applications

Laura Ricciotti, Alessio Occhicone, Claudio Ferone, Raffaele Cioffi, Oreste Tarallo and Giuseppina Roviello.

SUMMARY

The article submitted for review is devoted to a topical issue. It considers the sustainable design of materials based on geopolymers for art and restoration work. The authors examined the preparation processes and characteristics of the resulting solutions based on geopolymers obtained as a result of the processing of waste generated during the production process and the end of the service life of porcelain stoneware products.

The authors carried out large-scale structural, morphological, and mechanical studies and obtained several important results. Their study is quite deep from a scientific point of view, has a certain novelty, and is also practically significant.

Therefore, in general, the reviewer proposes to support this article. However, there were several comments that need to be corrected, they are presented below.

COMMENTS

1.    The authors presented the title in an applied style. The authors should have formulated the title more science-intensive due to the fact that scientific novelty can be traced quite clearly.

2.    In addition, the authors did not quite successfully present the abstract. It is not very clear what scientific problem the authors were solving? If they were trying to determine the most rational way to dispose and recycle porcelain stoneware, then this is one scientific problem. If they were aiming for a deep study in terms of structure and morphology to create new compounds, then this is another scientific problem.

3.    Authors should clearly state why their research was necessary and what kind of solution they are proposing: research or applied.

4.    In addition, the authors did not at all display the quantitative characteristics of the results obtained. It is not clear from the abstract how their study is superior to previous research results of other authors, that is, the abstract should be revised.

5.    The authors provide a review of the literature in the Introduction section. However, it is too small to talk about a specific scientific novelty. For example, in the paragraph from line 84 to line 91, 9 sources of literature are presented. At the same time, it is not clear which of these is different from the others, and by what criteria this difference is expressed. That is, the authors should indicate in more detail what was done earlier and formulate a scientific problem, scientific novelty, goals and tasks from this. This should logically follow from the literature review.

6.    In addition, the probably incorrect repetition of quoting in lines 35 and 38 is striking. Perhaps the authors should combine the first 2 paragraphs and put one common reference to the first source. If there is a typo here, then it should be corrected.

7.    I recommend adding a more detailed rationale for the selected materials in section 2.1 Materials.

8.    The description of the preparation of the mixture is not well presented.

9.    Perhaps the X-ray analysis in subsection 3.1 needs more explanation.

10.  Table 4 looks uninformative. Perhaps the authors should have presented it in another form?

11.  I recommend authors to add more detailed analysis of the SEM studies in Figures 3 and 4. This would allow the authors to deepen the studies in the structural and phase side.

12.  Figure 5 is poorly commented. It needs more detailed explanation.

13.  Figure 6 is presented in poor quality. Perhaps the authors have these image in a higher resolution?

14.  The Discussion is not presented in sufficient depth. Authors should give a detailed comparison of their results with those previously obtained by other authors.

15.  The Conclusions should be supplemented and specified in order to clearly identify the scientific result and prospects for the development of the study.

16.  In general, the article looks like a completed study, but needs to be finalized in accordance with the comments. It is also necessary to slightly correct the style of presentation and make some changes in the English language.

Author Response

Dear Reviewer,

we would like to thank you for the suggestions and comments that have allowed us to improve significantly and clarify some of the key topics addressed in the paper.

In the following we report the list of changes that we have done to the manuscript in order to address all the reviewers' comments. Please note that all changes have highlighted in the revised text by using the “track-change” mode.

  1. The authors presented the title in an applied style. The authors should have formulated the title more science-intensive due to the fact that scientific novelty can be traced quite clearly.

Authors' answer:

We thank the Reviewer for his/her comment. Title has been modified accordingly.

  1. In addition, the authors did not quite successfully present the abstract. It is not very clear what scientific problem the authors were solving? If they were trying to determine the most rational way to dispose and recycle porcelain stoneware, then this is one scientific problem. If they were aiming for a deep study in terms of structure and morphology to create new compounds, then this is another scientific problem.

Authors' answer:

We thank the Reviewer for his/her comment. Abstract has been modified accordingly. In particular it has been pointed out in a mor clear way that the authors in this study try to obtain geopolymer materials with good physical and mechanical properties for application in the field of art and design and restoration while, at the same time, valorizing industrial ceramic waste, thus lowering the environmental impacts of the new materials proposed.

  1. Authors should clearly state why their research was necessary and what kind of solution they are proposing: research or applied.

Authors' answer:

We thank the Reviewer for his/her comment. Several part of the manuscript have been modified accordingly. In detail, the authors have added some considerations to better explain the purpose of this research work: 1) the lack, to the best of our knowledge, of effective and environmental friendly materials based on geopolymers to be applied in the artistic and/or restoration sector. 2) the chemical and physical characterization of the proposed mortars, in which the use of large amounts of ceramic waste has been investigated to obtain low-cost and low environmental impact materials.

For this purpose, the authors have added information and considerations about the innovation and differences between the paper and other research on this topic in the Introduction (lines 69-77 and lines 104-120) and some other bibliographical references (9-14) have been added. Moreover, also Conclusions have been revised accordingly.

  1. In addition, the authors did not at all display the quantitative characteristics of the results obtained. It is not clear from the abstract how their study is superior to previous research results of other authors, that is, the abstract should be revised.

Authors' answer:

The abstract has been modified according to the Reviewer’s suggestion.

  1. The authors provide a review of the literature in the Introduction section. However, it is too small to talk about a specific scientific novelty. For example, in the paragraph from line 84 to line 91, 9 sources of literature are presented. At the same time, it is not clear which of these is different from the others, and by what criteria this difference is expressed. That is, the authors should indicate in more detail what was done earlier and formulate a scientific problem, scientific novelty, goals and tasks from this. This should logically follow from the literature review.

Authors' answer:

We thank the Reviewer for his/her comment. The Introduction and the Conclusion sections have been revise accordingly. In particular, the authors have added information and considerations to highlight the innovation of the research work, the objectives, and the difference with other works in the literature (lines 69-77 and lines 104-120), added some bibliographical references (9-14) and completely revised the Conclusions.

  1. In addition, the probably incorrect repetition of quoting in lines 35 and 38 is striking. Perhaps the authors should combine the first 2 paragraphs and put one common reference to the first source. If there is a typo here, then it should be corrected.

Authors' answer:

The mistake has been corrected according to the Reviewer’s suggestion.

  1. I recommend adding a more detailed rationale for the selected materials in section 2.1 Materials.

Authors' answer:

We thank the Reviewer for his/her comment. The authors have added details according to the Reviewer’s suggestion. (lines 145-148 in Experimental Section).

  1. The description of the preparation of the mixture is not well presented.

Authors' answer:

The authors have improved the section of sample preparation (2.2 Sample Preparation) according to the Reviewer’s suggestion.

  1. Perhaps the X-ray analysis in subsection 3.1 needs more explanation.

Authors' answer:

The authors have improved the section of the X-ray analysis in subsection 3.1 according to the Reviewer’s suggestion.

  1. Table 4 looks uninformative. Perhaps the authors should have presented it in another form?

Authors' answer:

We thank the Reviewer for his/her comment. Table 4 shows the chemical composition (weight %) of PS and RP used in this paper as obtained by EDX characterization. Errors in it have been corrected.

  1. I recommend authors to add more detailed analysis of the SEM studies in Figures 3 and 4. This would allow the authors to deepen the studies in the structural and phase side.

Authors' answer:

The authors have modified the text to improve comprehension (lines 188-201 and 214-16 in the Result and Discussion section).

  1. Figure 5 is poorly commented. It needs more detailed explanation.

Authors' answer:

The authors have added a description of Figure 5 (lines 287-292 in the Result and Discussion section).

  1. Figure 6 is presented in poor quality. Perhaps the authors have these image in a higher resolution?

Authors' answer:

The authors have improved the quality and resolution of Figure 6.

  1. The Discussion is not presented in sufficient depth. Authors should give a detailed comparison of their results with those previously obtained by other authors.

Authors' answer:

We thank the Reviewer for his/her comment. The authors, in order to deepen the discussion and better describe the innovation, objectives, and advancement from the data in the literature, made numerous detailed changes in the Abstract, Introduction, Results and Discussion, and finally in the Conclusion sections. All changes have been highlighted in review-mode in the text.

  1. The Conclusions should be supplemented and specified in order to clearly identify the scientific result and prospects for the development of the study.

Authors' answer:

The Conclusions have modified to identify the scientific result and prospects for the development of the study according to the Reviewer’s suggestion.

  1. In general, the article looks like a completed study, but needs to be finalized in accordance with the comments. It is also necessary to slightly correct the style of presentation and make some changes in the English language.

Authors' answer:

The article has been revised and modified in all sections according to the Reviewer suggestions. A complete revision of the English language has been made

Round 2

Reviewer 1 Report

In my previous report, I mentioned that the sum of all components of metakaolin listed in Table 1 gives 100,19% instead of 100%. Unfortunately, these data were not corrected in the version of the manuscript that I received.

P. 4, line 168. Please correct typo in the word “the”.

In my previous report, I mentioned that “…RC-MK where RC refers to raw pressed ceramic waste …” In Table 2, there is a different notation: “RP: raw pressed ceramic waste”. The authors answered “We thank the Reviewer for his/her comment. The mistakes have been corrected according to the Reviewer’s suggestion.” However, in the Manuscript version that I received the correction was not made.

In my previous report, I asked “9. Fig. 1A. Please discuss the XRD pattern of metakaolin. Why its crystallinity fraction is so low? Why there are no peaks of components constituting this material. There are only the peaks attributed to anatase, while the content of TiO2 in metakaolin is only 1.74 wt%.

Authors' answer:

We thank the Reviewer for his/her observation. In order to reply to this comment, we would like to point out that metakaolin consists mainly of an amorphous structure of silicon and aluminum oxides. This type of phase is widely visible in the amorphous halo with a center at 2θ = 23 degs. This is due to the heat treatment used to obtain the material from the starting kaolin. It is therefore not surprising that the only crystalline phase present is the anatase shown in the figure. Accordingly, no changes have been done on the manuscript.”

Does amorphous aluminum oxide exist not only in thin films?

In my previous report, I asked “12. In Figs 1 and 2, there are many peaks of sillimanite that are not marked with number 2. In the standard card of sillimanite, there are two main peaks located at 2theta = 25.9 deg. And 26.4 deg. with intensities of 81% and 100%, respectively. Such peaks are clearly seen in XRD patterns of PS and PRdry presented in Fig. 1 and in the XRD pattern of PS-MK? Fig. 2. The positions of peaks marked with number 1 are different in different XRD patterns of the materials probably because they are superimposed on the peaks of silimanite. In the XRD pattern of the RP materials, please mark all peaks denoted by the number 1.

Authors' answer:

We thank the Reviewer for his/her observation. Actually, we have not marked only very minor peaks just for the sake of readability of the graph.”

The authors did not make any correction of Figs 1 and 2. I cannot agree with their decision. The peaks of silimanite that they did not mark are not “very minor peaks”. They are the main peaks of silimanite.

The identification of crystalline phases presented in different patterns in Figs 1 and 2 causes questions because some patterns demonstrate the same peaks marked differently. Please, i.e., compare peaks marked with symbol “1” on Fig. 1B for RPdry and RP patterns. Please pay attention to the fact that the peak marked with symbol “1” is split only on the XRD pattern of RPdry. How do the authors attribute the second peak? Therefore I kindly ask the authors to mark the unmarked peaks and to mark with the same symbol only those peaks that have the same position on different figures.

Author Response

Reviewer Report, round 2

Development of geopolymer-based materials with ceramic waste for artistic and restoration applications

Naples, 25/11/2022

thank you again for the very careful review of the manuscript. We would like to apologize with Reviewer n.1 for the fact that some of his/her observation have not been completely addressed into the revision of the manuscript.

In the following we report the list of changes that we have done to the manuscript in order to address all the Reviewer' comments. Please note that all changes have highlighted in the revised text by using the “track-change” mode.

In my previous report, I mentioned that the sum of all components of metakaolin listed in Table 1 gives 100,19% instead of 100%. Unfortunately, these data were not corrected in the version of the manuscript that I received.

Authors' answer:

We apologize with the Reviewer. Data have been corrected, as requested.

  1. 4, line 168. Please correct typo in the word “the”.

Authors' answer:

We thank the reviewer for pointing out the typo, that we have corrected.

In my previous report, I mentioned that “…RC-MK where RC refers to raw pressed ceramic waste …” In Table 2, there is a different notation: “RP: raw pressed ceramic waste”. The authors answered “We thank the Reviewer for his/her comment. The mistakes have been corrected according to the Reviewer’s suggestion.” However, in the Manuscript version that I received the correction was not made.

Authors' answer:

We apologize with the Reviewer. Data have been corrected, as requested.

In my previous report, I asked “9. Fig. 1A. Please discuss the XRD pattern of metakaolin. Why its crystallinity fraction is so low? Why there are no peaks of components constituting this material. There are only the peaks attributed to anatase, while the content of TiO2 in metakaolin is only 1.74 wt%.

Authors' answer:

We thank the Reviewer for his/her observation. In order to reply to this comment, we would like to point out that metakaolin consists mainly of an amorphous structure of silicon and aluminum oxides. This type of phase is widely visible in the amorphous halo with a center at 2θ = 23 degs. This is due to the heat treatment used to obtain the material from the starting kaolin. It is therefore not surprising that the only crystalline phase present is the anatase shown in the figure. Accordingly, no changes have been done on the manuscript.”

Does amorphous aluminum oxide exist not only in thin films?

Authors' answer:

We apologize with the Reviewer for not being clear in our previous response. We would like to point out that, by stating “metakaolin consists mainly of an amorphous structure of silicon and aluminum oxides.”, we meant that the amorphous phase is composed of a disordered 3-D network made of silicon and aluminium atoms linked by oxygens, in line with what has been extensively described in literature to date (more details can be found for example in Davidovits J. Geopolymer Chemistry and Applications. Institut Géopolymère, Geopolymer Institute, Saint-Quentin, France Editor: Joseph Davidovits ISBN: 9782954453118 5th ed. (2020), ref. 48 of the manuscript). For this reason, in our reply, we were not referring to “amorphous aluminum oxide” as correctly pointed out by the Reviewer.

In my previous report, I asked “12. In Figs 1 and 2, there are many peaks of sillimanite that are not marked with number 2. In the standard card of sillimanite, there are two main peaks located at 2theta = 25.9 deg. And 26.4 deg. with intensities of 81% and 100%, respectively. Such peaks are clearly seen in XRD patterns of PS and PRdry presented in Fig. 1 and in the XRD pattern of PS-MK? Fig. 2. The positions of peaks marked with number 1 are different in different XRD patterns of the materials probably because they are superimposed on the peaks of silimanite. In the XRD pattern of the RP materials, please mark all peaks denoted by the number 1.

Authors' answer:

We thank the Reviewer for his/her observation. Actually, we have not marked only very minor peaks just for the sake of readability of the graph.”

The authors did not make any correction of Figs 1 and 2. I cannot agree with their decision. The peaks of silimanite that they did not mark are not “very minor peaks”. They are the main peaks of silimanite.

Authors' answer:

We apologize with the Reviewer for the mistake. In this new revision, following the reviewer's instructions, all phases were re-evaluated, and all the numbers have been reassigned, correcting inconsistencies correctly reported by the Reviewer. For these reasons, the figures have been changes following the suggestions and comments made by the reviewer.

The identification of crystalline phases presented in different patterns in Figs 1 and 2 causes questions because some patterns demonstrate the same peaks marked differently. Please, i.e., compare peaks marked with symbol “1” on Fig. 1B for RPdry and RP patterns. Please pay attention to the fact that the peak marked with symbol “1” is split only on the XRD pattern of RPdry. How do the authors attribute the second peak? Therefore I kindly ask the authors to mark the unmarked peaks and to mark with the same symbol only those peaks that have the same position on different figures.

Authors' answer:

We apologize with the Reviewer for our mistake. Following the reviewer's instructions, both figure 1 and figure 2 have been modified according to his/her suggestions.

Finally, some minor typos have been corrected,

Best Regards,

Laura Ricciotti

Giuseppina Roviello

Reviewer 2 Report

Ok

Author Response

No request was received from the Reviewer

Reviewer 3 Report

all my comments were taken into account and appropriate corrections were done in the text. 

I recommend the article for publishing.

Author Response

(The authors gave the same response as above.)
